# Link Prediction Based on Deep Convolutional Neural Network

**Wentao Wang [1], Lintao Wu [1,\*], Ye Huang [1], Hao Wang [2] and Rongbo Zhu [1]**

[1] College of Computer Science, South-Central University for Nationalities, Wuhan 430074, China; wangwt@mail.scuec.edu.cn (W.W.); 2016110265@mail.scuec.edu.cn (Y.H.); rbzhu@mail.scuec.edu.cn (R.Z.)

[2] College of Computer Science and Technology, Chongqing Engineering Research Center for Spatial Big Data Intelligent Technology, Chongqing University of Posts and Telecommunications, Chongqing 400065, China; haowang@cqupt.edu.cn

\* Correspondence: 2017110246@mail.scuec.edu.cn

**Abstract:** In recent years, endless link prediction algorithms based on network representation learning have emerged. Network representation learning mainly constructs feature vectors by capturing the neighborhood structure information of network nodes for link prediction. However, this type of algorithm only focuses on learning topology information from the simple neighbor network node. For example, DeepWalk takes a random walk path as the neighborhood of nodes. In addition, such algorithms only take advantage of the potential features of nodes, but the explicit features of nodes play a good role in link prediction. In this paper, a link prediction method based on deep convolutional neural network is proposed. It constructs a model of the residual attention network to capture the link structure features from the sub-graph. Further study finds that the information flow transmission efficiency of the residual attention mechanism was not high, so a densely convolutional neural network model was proposed for link prediction. We evaluate our proposed method on four published data sets. The results show that our method is better than several other benchmark algorithms on link prediction.

**Keywords:** link prediction; network representation learning; deep learning; residual network; attention mechanism

## 1. Introduction

Complex systems in the real world can usually be constructed in the form of networks with nodes representing different entities in the system and links representing the relationships between these entities. Link prediction is to predict whether two nodes in a network are likely to have a link [1]. It is used for diverse applications such as friend suggestion [2], recommendation systems [3,4], biological networks [5] and knowledge graph completion [6].

Existing link prediction methods can be classified in the similarity-based method and the learning-based method. The similarity-based method assumes that the more similar the nodes are, the greater the possibility of links are between them [7,8]. It calculates the similarity between nodes by defining a function that can use some network information, such as network topology or node attributes, to calculate the similarity between nodes, and then utilize the similarity between nodes to predict the possibility of links between nodes. The accuracy of prediction largely depends on whether the network structure features can be selected well or not. The learning-based method constructs a model that can extract various features to build a model for the given network, train the model with the existing information, and finally use the trained model to predict whether there will be links between the nodes.

In early works, heuristic scores are used for measuring the proximity of connectivity of two nodes in link prediction [1,9]. Popular heuristic methods include the similarity method based on local information and the similarity method based on path similarity. The similarity methods based on local information only considers the local structure, such as the common neighbors of two nodes, as a measure of similarity [1,10]. The method based on path similarity utilizes the global structural information of the networks, including paths [11,12] and communities [13], as a basis to find the similarity of nodes. However, the structure-based method entirely relies on the topology of the given network. Furthermore, the structure-based method is not that reliable, and there is such a situation where different networks may have distinct clusterings and path lengths but have the same degree distributions [14]. Therefore, it can only show the different performance for each network and is unable to effectively capture the underlying topological relationship between nodes.

In recent years, more and more learning-based algorithms have been proposed. This method mainly extracts the features of the network by constructing a model and then predicts the links through the models. Learning-based methods can be divided into two categories: shallow neural network-based method and deep neural network-based method. DeepWalk [15] is an algorithm based on the shallow neural network. It treats the path of a random walk as a sentence to learn the representation vectors of nodes through the skip-gram model [16], which greatly improves the performance of the network analysis task. The LINE (Large-scale Information Network Embedding) [17] algorithm based on shallow neural network and node2vec [16] algorithm based on modified DeepWalk then occur successively. Since the shallow model does not capture the highly nonlinear network structure, which results in the non-optimal network representation results, a semi-supervised depth model based on the SDNE (Structural Deep Network Embedding) [18] deep neural network algorithm, which is composed of multi-layer non-linear functions, is put up to capture highly non-linear network structure. However, these two methods only take advantage of the potential features and cannot effectively capture the structural similarity of links [19].

Depth neural network has made great progress in image classification, target detection and recognition in recent years because of its powerful feature learning and expression abilities [20]. However, deep neural network has the problem of gradient disappearance when deepening the depth [21]. The residual network [21] mentioned in CVPR2016 (CVPR2016: IEEE Conference on Computer Vision and Pattern Recognition) has not only achieved good results in image recognition and target detection tasks but also solved the degradation problem of network learning ability caused by network deepening [21]. The residual network is used to further deepen the number of network layers by introducing an identity mapping [22] into the original network structure. Adding such a short connection essentially reduces the loss of information between network layers. Dense convolutional neural network [23] is an improved version of the residual network. By introducing more short connections, information flow can be transmitted more effectively in the network, thereby achieving better recognition and detection results. Therefore, this paper proposes a link prediction method based on deep convolutional neural network to predict missing/unknown links in the network.

To address the above-mentioned problems, we propose a link prediction method based on the deep convolution neural network. The main contributions of our work can be summarized as follows:

1. To solve the link prediction problem, we transform it into a binary classification problem and construct a deep convolution neural network model to solve the problem.
2. In view of the fact that heuristic methods can only utilize the network's topological structure and represent learning methods can only utilize the potential features of the network, such as DeepWalk, LINE, node2vec, we propose a sub-graph extraction algorithm, which can better contain the information needed by the link prediction algorithm. On this basis, a residual attention model is proposed, which can effectively learn from graph structure features to link structure features.

3.     Through further research, we find that the residual attention mechanism may impede the information flow in the whole network. Therefore, a dense convolutional neural network model is proposed to improve the effect of link prediction.

The remainder of the paper is organized as it follows. In the next section, the related works are presented. In Section 3 the preliminaries about this paper are introduced. In Section 4 the sub-graph extraction algorithm and residual attention network are proposed. The performance evaluation results and discussion are summarized in Section 5, while conclusive remarks are given in the last section.

## 2. Related Work

In general, the deeper the neural network is, the more it learns. However, simply superimposing the number of network layers will lead to the problem of network degradation [21], the essential reason of which is that in the process of network information transmission, there is an over-fitting problem caused by information loss. By adding a "short cut" design into the original network architecture, the residual network makes the data of the previous layers directly skip the next layers and act as the input part of the latter data layer, so that with the reference of the former data information, the latter network layers can learn more information. However, because the output of the network front layer and "short cut" are firstly integrated by adding and then as the input of the next layer, this integrated information will actually hinder the dissemination of information on the whole network [23].

In recent years, many researchers have integrated the attention mechanism into the residual network in the field of image recognition. As some pictures have complex backgrounds and complex environments, it is necessary to pay different attention to different places. Therefore, this paper, inspired by the attention mechanism network [24], also adds a branch of attention on the structure of the residual network by simulating the design of "short cut." Several layers in the front of the network transformed by non-linearity and attention mechanism will be added with the "short cut" to form the input of the latter layer. The problem of this method is that the information after integration will hinder the transmission of information on the whole network. In order to improve the propagation of information flow in the network, Huang [23] proposed a dense convolutional neural network which takes the features of the front layers directly as the input of the back layers by a deep cascade, i.e., combining channels in depth, so as to effectively improve the transmission of information flow in the network.

The main limitation of traditional similarity-based methods is that all their features are designed by hand, which limits the scalability, so they cannot express complex non-linear patterns in graphs. Based on the shallow neural network model, they only take advantage of some potential features, and cannot effectively extract the structural features of links. Therefore, this paper proposes a link prediction method based on a deep convolution neural network, which uses deep learning to learn link structure from a graph, and finally achieves good results.

## 3. Preliminaries

### 3.1. Network Representation Learning

Network Representation Learning [25,26] has been widely used since the pioneering work of DeepWalk which uses random walks as node sentences and applies skip-gram models to learn the node representation vector. LINE [17] and node2vec [16] are proposed to improve DeepWalk. The low-dimensional node representation vector is very helpful for visualization, node classification, link prediction and other tasks. In particular, node2vec has achieved very good results in link prediction [16].

Representation learning based on Skip-gram model was originally used as a representation vector of learning words. The objective of Skip-gram model is to maximize the probability of the occurrence

of the current word in the context, while the optimization objective of the network representation learning is to maximize the probability of neighbor nodes in the target node of the node sequence.

$$\max_f \sum_{u \in V} \log P(N(u)|f(u)),\qquad(1)$$

where $V$ denotes a set of network node; $N(u)$ denotes the set of neighbor nodes of the node $u$ in the node sequence; $f(\cdot)$ denotes the mapping relation between nodes and feature space; and $f(u)$ denotes the corresponding feature representation vector of the node $u$.

Compared with other network representation learning algorithm, node2vec has stronger scalability and expressibility. Therefore, we chose the representation vector learned from node2vec algorithm as the potential feature representation vector of nodes in this paper.

### 3.2. Residual Attention Mechanism

For the ordinary convolutional neural network and it's arbitrarily stacked two-layer network, as shown in Figure 1, we hoped to find the residual element of the expected mapping $H(x)$, that is, $x$ obtains an actual observation value through non-linear change, and there is a residual element between the actual observation value and the estimated value (i.e., the expected value).

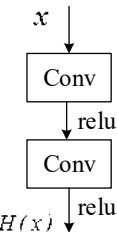

**Figure 1.** Plain convolutional neural network block.

He et al. [21] add a shortcut connection $x$ to the ordinary network architecture, as shown in Figure 2, so that the residual is described as $R(x) := H(x) - x$. With the assumption that the residuals are more easily optimized and fit the residuals mapping with stacked non-linear transformations, the expected mapping turns to $H(x) = R(x) + x$, and the problem changes from looking for $H(x)$ to $R(x)$. Among them, $x$ and $R(x)$ require the same size (dimensions must be equal); if their sizes are different, a linear map $wx$ is required to make $x$ and $R(x)$ the same size.

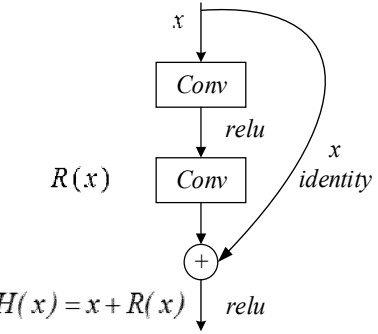

**Figure 2.** The classical residual block structure.

In recent years, the attention model has been widely used in various types of in-depth learning tasks, such as natural language processing, image recognition and speech recognition. Wang et al. [24] mentioned that the attention mechanism used in the deep convolutional neural network can not only help feature selection in forward inference but also act as gradient update filter in backward

propagation [24], which makes the attention mechanism robust to noise labels. Therefore, we add a branch of attention mechanism to the structure of the residual network as the model of this paper.

### *3.3. Densely Connected Convolutional Neural Network*

Although the residual attention mechanism has achieved good results in the experiment, it is found that the model still fails to effectively solve the problem of information flow propagation in the network.

In order to further improve the transmission of information flow in the network, Huang et al. [23] proposed a densely convolutional neural network model. In fact, the densely convolutional neural network is a special case of the residual network, the core idea of which is skipping connection. It allows some inputs to enter the layers without selection, thus realizing the integration of information flow and avoiding the loss of information transmission between layers and the disappearance of gradient. Compared with the residual network, which only introduces a single "shortcut connection," the densely convolutional neural network introduces more "shortcut connections," as shown in Figure 3, and this reference information is not processed, but directly cascaded in depth, that is, channel merging in depth.

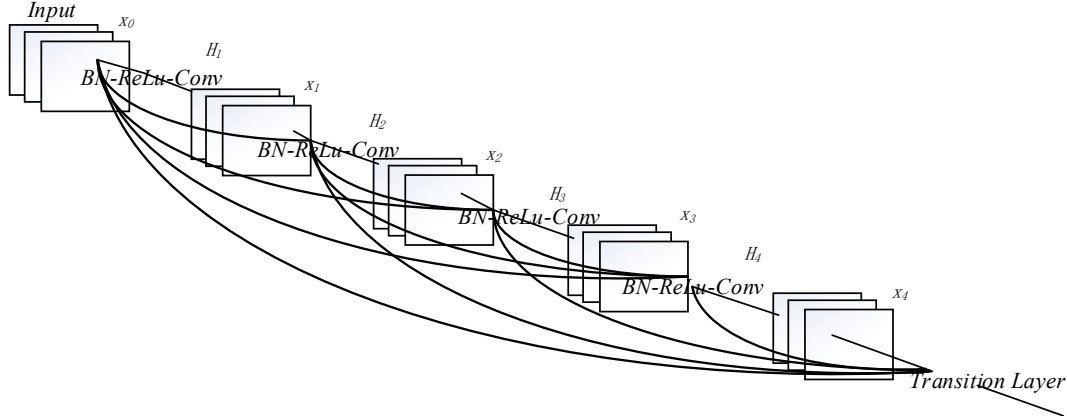

**Figure 3.** A five-layer dense block with a growth rate of $k = 3$. Where $k$ denotes the number of feature map output from each layer of the network, in this diagram, we draw three.

## 4. Methodology

### *4.1. Problem Formulation*

Like most works on link prediction denoted, we considered an unweighted and undirected network $G = (V, E)$. $V$ and $E$ represent the sets of nodes and links in the network respectively. The adjacency matrix of the network is denoted as $A$, if nodes $i$ and $j$ have a link, then $A_{ij} = 1$; otherwise $A_{ij} = 0$.

In order to simulate link prediction tasks, 10% of links are randomly removed from the original network $G = (V, E)$ and recorded as $E^p$ and it shall ensure network connectivity while removing links. A new network $G' = (V, E')$ is generated where $E^p \cap E' = \varnothing$, $E'$, $E^p$ are positive classes in the training set and test set respectively. Then, random additions and positive class equivalents without edges as negative classes. When negative classes are added, the intersection of the training set and test set is ensured to be empty.

### *4.2. Method Overview*

The problem of link prediction in a network is simply to predict the possibility of links between two nodes in a network that have not yet been connected by known or potential information in the network. In other words, the information in the network can be used to determine whether two nodes

will be connected. Formally, the link prediction problem is transformed into a binary classification problem, and the classification problem is solved by constructing a learning model. Specifically, in data pre-processing, we first extracted a sub-graph for each node of the network, and then sorted the sequence of nodes in the sub-graph. Finally, we reconstructed the ordered sequence of nodes into matrix pairs (one node corresponds to a matrix, one pair of nodes corresponds to two matrices). In practice, we first built a two-class residual attention network model to achieve link prediction. After further research, we found that the densely convolutional neural network can further improve the information flow from layers, so we adopted it.

### 4.3. Sub-Graph Extraction

A significant limitation of link prediction heuristics is that they are all handcrafted features, which have limited expressibility. Thus, they may fail to express some complex nonlinear patterns in the graph which actually determine the link formations [19]. In order to learn link structure features, we extracted a corresponding local sub-graph for each node to obtain the local structure of each node. The sub-graph extraction algorithm is given by Algorithm 1.

**Definition 1 (sub-graph):** *Given undirected graph $G = (V, E)$, where $V = \{v_1, v_2, \ldots, v_n\}$ is the set of nodes and $E \subseteq V \times V$ is the set of observed links. For node $x \subseteq V$, the h-hop sub-graph for x is $G_x^h$ induced from G by the set of nodes $\{y | d(y, x) \leq h\}$ and its corresponding nodes and links. The $d(y, x)$ is the shortest path distance between x and y.*

---

**Algorithm 1** Sub-graph extraction algorithm

---

*input:* Objective node $x$, network G, integer $h$
*output:* $x$ corresponding sub-graph $G_x^h$
1. $V_x^h = \{x\}$
2. $curr\_nb = \{x\}$
3. **for** $i$ in range(h)
4.　　 **if** $|curr\_nb| == 0$ then break
5.　　　 $curr\_nb = (\cup_{v \in curr\_nb} N(v)) \backslash V_x^h$
6. $V_x^h = V_x^h \cup curr\_nb$
7. **end for**
8. **return** $G_x^h = G(V_x^h)$

---

where the $h$ denotes the hop number contained in the sub-graph; $N_{(v)}$ is the 1-hop neighbors of $x$; $G(\cdot)$ represents a sub-graph containing a set of target nodes and is generated from the original graph.

Network robustness depends on several factors including network topology, attack mode, sampling method and the amount of data missing, generalizing some well-known robustness principles of complex networks [27]. Our proposed sub-graph extraction algorithm is mainly used for sampling, so we discuss and care about the robustness of sampling.

The sub-graph extraction algorithm describes the h-hop surrounding environment of node $x$, since $G(\cdot)$ contains all the nodes within $h$ hops to $x$ and corresponding edges. For example, when $h \geq 2$, the extracted sub-graph will contain all the information needed to calculate the first-order and second heuristics algorithms, such as CN (Common Neighbors) [10], PA (Preferential Attachment) [28] and AA (Adamic-Adar) [29]. However, this kind of algorithm only considers the topology of the second-order path, the time complexity is low, but the prediction effect is also poor. Therefore, the information obtained from the sub-graph is at least as good as most heuristics, and the robustness is also strong.

### 4.4. Node Sorting

The key issue for applying the convolutional neural network into the network embedding is how to define the receptive field on the network structure. Therefore, the first step of the algorithm is to take the local structure of each node as the input of the receptive field. The way to obtain the local

structure of the node is described in Algorithm 1. The convolution operator has been shown to be effective in exploiting the correlation as the key contributor to the success of CNNs (Convolution Neural Networks) on a variety of tasks [30]. However, for the network topology data form, which is irregular and unordered, the convolution operator is ill-suited for leveraging spatially-local correlations in the network structure [31]. Therefore, inspired by this, the second step of the algorithm is to transform the local structure of each node into an ordered sequence.

To sort the nodes in the sub-graph, the first step is to generate representation vectors for each of nodes in the network by the node2vec algorithm; the second step is to calculate the similarity scores between the target node and other nodes in the sub-graph; the last step is to descend sequence of nodes according to similarity score. It supposes that $X = [x_1, x_2, \ldots, x_d]$ represents the $d$-dimensional vector representation of any node $x$ and $Y = [y_1, y_2, \ldots, y_d]$ represents the $d$-dimensional vector representation of any node $y$. Since the cosine distance is widely used to measure the correlation between two points in multi-dimensional space, this paper also uses the cosine distance between the nodes in multidimensional space to represent the similarity of each node in the network structure. Therefore, the nodes are sorted according to the similarity. The node sorting algorithm is given by Algorithm 2.

**Definition 2:** *The similarity of network structure between any nodes x and y is as follows:*

$$\text{sim}(X, Y) = \cos(X, Y) = \frac{\sum\limits_{i=1}^{d} x_i y_i}{\sqrt{\sum\limits_{i=1}^{d} x_i^2 |\bullet|} \sqrt{\sum\limits_{i=1}^{d} y_i^2}}, \tag{2}$$

*where X, Y represent the representation vector of the node x, y, respectively, and the subscript i represents the dimension, for a total of d dimensions. The similarity is measured by calculating the cosine distance between them. The higher the vector similarity of the two nodes, the closer the two nodes are, so they should be in the first place when ranking.*

---

**Algorithm 2** Node sorting algorithm

---

*input: nbb[n], node_vec[N][M]*
*output: x*
1. *node_vec_dist*[][];
2. **for** $i \leftarrow 1$ to $n$ **do**
*node_vec_dist[nbb[0]][nbb[i]]*=cos(*node_vec[nbb[0]],node_vec[nbb[i]]*);
3. **end for**
4. seq = Sorted(node_vec_dist, Reverse = True)
5. **return** seq

---

Table 1 shows the meaning of the variable names in Algorithm 2.

**Table 1.** Meanings of the variable names in the node sorting algorithm.

| Variable Name | Meanings |
|---|---|
| *n* | Number of nodes in sub-graph |
| *N* | Number of nodes in network G |
| *M* | Represents the dimension of feature |
| *Nbb*[] | *x* corresponding ordered node sequence *seq[n]* |
| *node_vec*[][] | All nodes in the network represent vector |
| *node_vec_dist*[][] | A temporary variable that holds the cosine distance |

### 4.5. Node Information Matrix Construction

Link prediction aims to predict the relationship between two nodes. The most intuitive assumption is that the more similar the two nodes are, the more likely they are to be connected. Therefore, this paper considers the two nodes as a whole directly. In this paper, we encode the corresponding sub-graph of the nodes as a $k \times k \times 2$ size adjacency matrix, where 2 represents two nodes. The overall meaning is that two matrices are cascaded in depth. The adjacency matrix $A$ referred to below is the matrix mentioned in Section 4.1.

The node information matrix for a sub-graph (as shown in Figure 4) is constructed by the following steps:

(1) For a given graph $G = (V, E)$, corresponding h-hop sub-graph for each node shall be generated according to Algorithm 1 Sub-graph extraction algorithm;

(2) The nodes in each sub-graph shall be sorted, and the order between sub-graphs does not affect each other;

(3) If the number of nodes in the sub-graph is greater than or equal to $k$, the first $k$ nodes are selected from the ordered sequence nodes, and the $k$-ordered sequence nodes is mapped to the adjacency matrix $A_x^k$, $A_y^k$; If the number of nodes is less than $k$, the element 0 shall be filled as complementary element;

(4) $A_x^k$ and $A_y^k$ are integrated into data with the size of $k \times k \times 2$. If the node pair $(x, y)$ has a link in the original network, it is labeled as 1; otherwise, it is labeled 0.

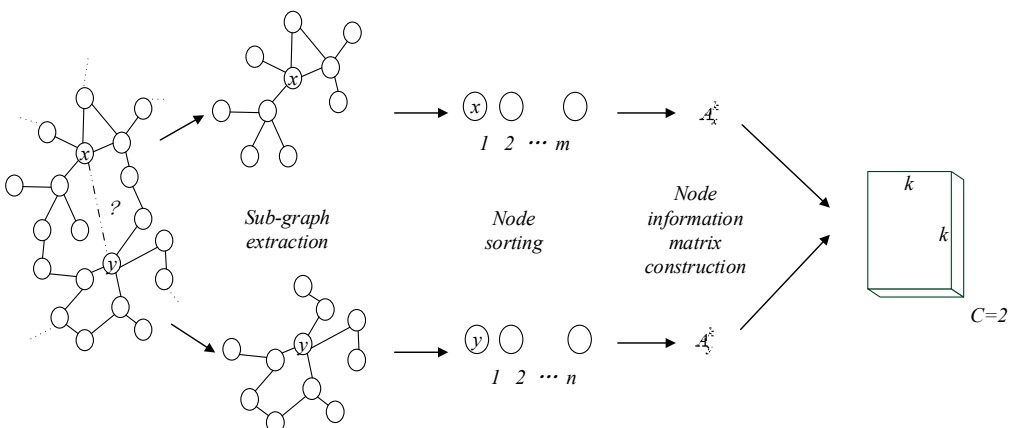

**Figure 4.** The construction process of the node information matrix for a sub-graph.

### 4.6. Residual Attention Mechanism

The residual network based on attention mechanism in this paper is composed of stacks of residual attention modules. The residual attention module (Figure 2) is formed by adding an attention mechanism branch to the classic residual block (Figure 5). Wang [24] proposed that attention branches can not only serve as a feature selector during forwarding inference but also as a gradient update filter during backpropagation.

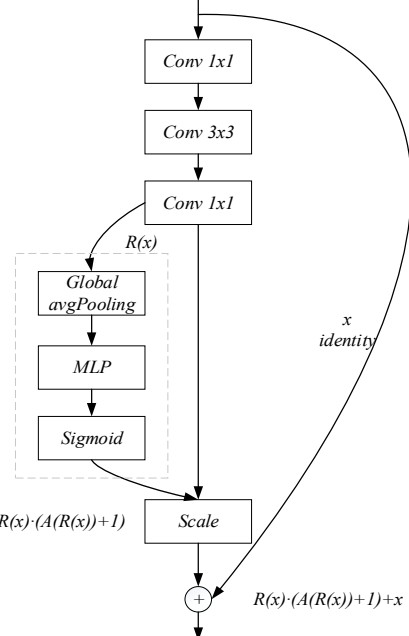

**Figure 5.** Residual attention module.

In the attention mechanism branch, the output of $R(x)$ after passing the residual block is assumed to be $N \times W \times H \times C$, and $N$ of which is the number of input data, $W \times H$ the size of input data, and $C$ the convolution channel. Firstly, the output of the residual block $R(x)$ generates the average pooling of channel characteristics. The form is defined as follows:

$$G(x)_c = \frac{1}{W \times H} \sum_{i=1}^{W} \sum_{j=1}^{H} R(x)_c(i, j). \tag{3}$$

The subscript $c$ indicates the calculated channel.

The feature weight of the adaptive learning channel is obtained by the feature redirection of the multi-layer perceptron (MLP). The channel attention mask A with the size of $N \times 1 \times 1 \times C$ is obtained through the sigmoid activation. The calculation formula is given in Equation (4), and its value is normalized to the range of (0, 1). The larger the value, the more important the corresponding channel characteristics.

$$A = \sigma(f(\{W_i\}, G)). \tag{4}$$

$G$ is the channel characteristic weight distribution calculated by Equation (3); $f(\{W_i\}, G)$ denotes the calculation of MLP; and $\sigma$ is the sigmoid function. Therefore, $A$ is clarified into (0, 1). Inspired by residual learning, if the attention unit can form the identical mapping, its performances should be no worse than its counterpart without attention. Thus, we modify the model $H(x)$ into:

$$H(x) = R(x) \bullet (A(R(x)) + 1) + x, \tag{5}$$

where $\bullet$ is a dot production operation and $A(R(x))$ is used to represent the channel attention mask $A$. When $A$ approximates 0, $H(x)$ will approximate the original feature $R(x)$. For ease of exposition, the above approach is named AM-ResNet-LP.

*4.7. Densely Connected Convolutional Network Model*

Residual network adds a short cut link connection, that is, adds an identity mapping bypass on the basis of non-linear transformation:

$$x_l = x_{l-1} + H_l(x_{l-1}),\tag{6}$$

where $H_l(\bullet)$ represents a non-linear transformation and $l$ indexes the layer. By adding such an identity mapping in the front layer and back layer of the network, the problem of gradient disappearance has been greatly reduced. However, the identity mapping and non-linear transformation are combined by addition, which may hinder the effective transmission of information flow in the network [23]. Similarly, the problem also exists in the residual attention mechanism.

To further improve the information flow between layers, G. Huang [23] propose a new network architecture, DenseNet, which is a more radical dense connection mechanism. The $l$th layer of the network receives the feature-maps of all preceding layers, $x_l = H_l([x_0, x_1, \ldots, x_{l-1}])$ and $[x_0, x_1, \ldots, x_{l-1}]$ refers to the concatenation of the feature-maps produced in layers $0, \ldots, l-1$, that is, the channel is merged in depth. For ease of exposition, the above approach is named DenseNet-LP.

## 5. Experimental Results

In this section, the proposed methods have conducted experiments in four real social network data sets and been compared with the classic and current methods or link prediction. These methods will be introduced in Section 5.2.2. The results show that the methods adopted have superiority and robustness for link prediction and performed well on various networks. To evaluate these methods, four different data sets including USAir line, PoliticalBlogs, Metabolic, and King James are employed, which are introduced in Section 5.1. The experiment results will be discussed in detail in Section 5.3.

*5.1. Data Sets*

Four networks from the social field, biological field and information field were employed to evaluate the performance of the method proposed method in this paper. The networks used in the experiment are described as follows and the basic statistical features are shown in Table 2. Directed links are treated as undirected; multiple links are treated as a single unweighted link and the self-loops are removed. The experimental data involved in this article are all from real networks, so readers can download them on the websites (http://www.linkprediction.org/index.php/link/resource/data/).

(1) USAir line (USAir): The air transportation network of USA that consists of 332 nodes and 2126 links. The nodes of the network are airports. If there is a direct route between two airports, then there is a link between the two airports.

(2) Politicablogs (PB): The network of American political blog website that consists of 1222 nodes and 19,021 links. The nodes of the network are log pages, and each link represents the hyperlinks between the blog pages.

(3) Metabolic: A metabolic network of nematode that consists of 453 nodes and 2025 links. The node of the network is metabolite and each link represents a biochemical reaction.

(4) King James: A vocabulary co-occurrence network that consists of 1773 nodes and 9391 links. The nodes of the network represent nouns and the links indicate that two nouns appear in the same sentence.

**Table 2.** The basic topological features of the networks.

| Dataset | \|V\| | \|E\| | <k> | <CC> | r |
|---------|------|-------|------|------|------|
| USAir | 332 | 2126 | 12.810 | 0.749 | −0.208 |
| PB | 1222 | 16,714 | 27.360 | 0.360 | −0.221 |
| Metabolic | 453 | 2025 | 8.940 | 0.647 | 0.226 |
| King James | 1773 | 9131 | 18.501 | 0.163 | −0.0489 |

Table 2 shows the basic topological features of four real networks studied in this paper. |V| and |E| are the numbers of nodes and links; <k> is the average degree; CC is the clustering coefficient; r is the assortative coefficient.

To evaluate the accuracy of the proposed method with others, we used the AUC (Area Under the receiver operating characteristic Curve) score as the index which can be interpreted as the probability that the value of a randomly chosen fraction with links is higher than that of a random fraction without links [9]. In general, the value of AUC will be between 0.5 and 1 and the higher the value of AUC, the higher the algorithm accuracy and the maximum is. Formulaic definitions are as follows:

$$AUC = \frac{N' + 0.5N''}{N},\tag{7}$$

where $N$ is the number of independent repeated times; $N'$ is the number of times that the score with links in the test set is higher than that without links, and $N''$ is the number of times that the score with links in the test set is equal to that without links.

### 5.2. Experiment Setup

#### 5.2.1. Parameter Settings

(1) To quantify the performance of the link prediction, the presented links from each data set were partitioned into a training set (90%) and test set (10%) randomly and independently. At the same time, the network connectivity of the training set and the test set was guaranteed.

(2) To compare the results with other learning methods, the same parameter settings were adopted. Please refer to [20] for details. In addition, the node representation vectors generated by the two models were computed according to Equation (8), and the link eigenvectors were computed by the Hadamard [20] operation mentioned. The calculation formula is as follows:

$$F_{(u,v)} = [f(u) \otimes f(v)]_i = f_i(u) * f_i(v),\tag{8}$$

where $f(u)$ denotes the representation vector of the node $u$; $f_i(u)$ denotes the value of the $i$-dimension of the representation. Binary regression classifier is used to predict unknown links.

(3) Sub-graph extraction algorithm: it is usually set as $h \in \{2,3\}$. Since $h = 3$ achieved good results in the experiment, we set $h = 3$.

(4) Node information matrix construction: the experiment sets $k \in \{32, 64, 128\}$. Through the experiment contrast, when $k = 64$, the prediction accuracy of AUC is relatively good, so we chose the size of $64 \times 64$, as shown in Figure 6.

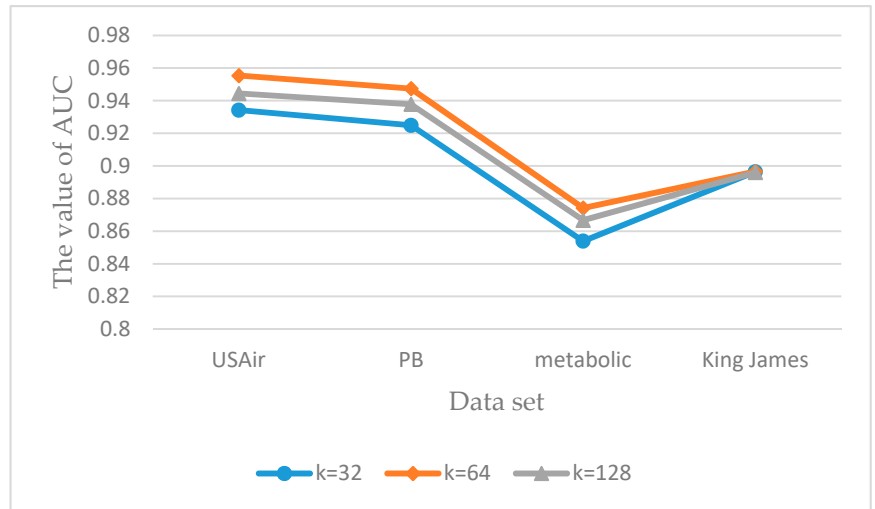

**Figure 6.** The effect of the AUC value on model DenseNet with parameter *k*.

### 5.2.2. Baseline Algorithms

This method is compared with several well-known methods, including learning methods based on CN, Jaccard, AA, PA index and other baselines. These methods are denoted as following, respectively.

DeepWalk [15]: DeepWalk is the first algorithm for network embedding, which uses Word2Vec model to learn structural feature vectors. In DeepWalk, the communities of network are disregarded during the path generation

Node2vec [16]: The representation learning process of this model is similar to DeepWalk. However, it employs a more flexible definition of neighborhood to facilitate random walks. Node2vec ignores the community information and higher order proximities during path generation.

LINE [17]: In the LINE algorithm, the node's feature vectors are generated by optimizing two independent functions for the first and second order proximities. Then, combinations of two functions are employed to provide final structural feature vectors. LINE also neglects the community information of network topology.

### 5.3. Experiment Results and Analysis

These methods mentioned above were experimented on four real public data sets. In order to present the prediction results more accurately, the independent experiments were repeated 20 times on all data sets, and the AUC average of these 20 experiments was calculated as the final prediction results. Table 3 is a comparison with the CN-based methods. Table 4 is a comparison with the network representation learning algorithm.

**Table 3.** Comparison with CN-based methods (AUC).

| Data Set | CN | AA | Jac | PA | AM-ResNet-LP | DenseNet-LP |
|----------|------|------|------|------|--------------|-------------|
| USAir | 0.9368 | 0.9507 | 0.9072 | 0.8876 | 0.9550 | 0.9554 |
| Metabolic | 0.8448 | 0.8540 | 0.8050 | 0.8229 | 0.8632 | 0.8742 |
| PB | 0.9218 | 0.9250 | 0.8760 | 0.9022 | 0.9322 | 0.9474 |
| King James | 0.6543 | 0.6690 | 0.6621 | 0.5195 | 0.8877 | 0.8966 |

**Table 4.** Comparison with network embedding algorithms (AUC).

| Data Set | LINE | DeepWalk | node2vec | AM-ResNet-LP | DenseNet-LP |
|----------|------|----------|----------|--------------|-------------|
| USAir | 0.8066 | 0.7665 | 0.8349 | 0.9550 | 0.9554 |
| Metabolic | 0.7733 | 0.7871 | 0.7970 | 0.8632 | 0.8742 |
| PB | 0.7779 | 0.7783 | 0.7911 | 0.9322 | 0.9474 |
| King James | 0.8634 | 0.8602 | 0.8895 | 0.8877 | 0.8966 |

### 5.3.1. Comparison with CN-Based Methods

As shown in Table 3, the algorithm adopted in this paper was compared with some local neighbor-based algorithms for the link prediction, among which, DenseNet-LP is the best link prediction method. Firstly, it can be found that sub-graph-based methods generally have better performance than all CN-based methods, which demonstrates the advantage of learning over handcrafting graph features. The CN-based algorithms only focus on the first order proximities. While we consider higher order proximities in our method. In addition, the methods of CN and AA achieve good performance on the data sets of the USAir and PB, because the average degree, clustering coefficient and other network properties are relatively high, which indicates that the degree of tightness between nodes is relatively high. Therefore, it proves that this method can achieve better results. However, when the clustering coefficient of the network is very small or the assortative coefficient is also low, the degree of tightness between nodes and the prediction value of the AUC is very low, for instance, King James. It shows that the prediction effect of CN-based method is closely related to the attributes of network structure and the generalization performance of this method is relatively weak.

### 5.3.2. Comparison with Representation Learning Methods

In Table 4, different structural feature vectors can be obtained by the baseline network representation learning algorithms. As shown in Table 4, AM-ResNet-LP achieves better performance in contrast to other methods because both latent and explicit features of the network are employed in our method. Node2vec has a better performance in comparison to DeepWalk, because DeepWalk's sampling strategies for nodes is random, resulting in different learned feature representations. While in node2vec, it overcomes the problem of inflexible sampling of nodes from a network by designing a flexible objective that is not tied to a particular sampling strategy and providing parameters to tune the explored search space. Node2vec is able to capture more general information from the graph but unable to capture the structural similarities of links.

In order to further illustrate the experimental results, we calculated the confidence interval of the classification accuracy predicted by the proposed method. As we regard link prediction a binary decision, which may be correct or wrong, we assume that it is general. We are interested in a 95% confidence interval. The formula is as follows:

$$Pr(c1 <= \mu <= c2) = 1 - \alpha, \tag{9}$$

where $\alpha$ is the significant level (we chose 0.05), $\mu$ represents the average value of AUC, $Pr$ is the abbreviation of probability.

We calculated the confidence intervals of the predicted results of the two methods, in which the sample we took was the size of the test data set. As can be seen from Table 5, the confidence interval ranges from 1.5% to 7%, which shows that our experimental results are credible.

**Table 5.** Confidence interval of classification accuracy.

| Model | Dataset | Simple Size | AUC | Interval (c1,c2) |
|---|---|---|---|---|
| DenseNet-LP | USAir | 424 | 0.9554 | 0.935~0.975 |
| | Metabolic | 404 | 0.8742 | 0.836~0.902 |
| | PB | 3342 | 0.9474 | 0.940~0.955 |
| | King James | 1810 | 0.8966 | 0.882~0.910 |
| AM-ResNet-LP | USAir | 424 | 0.9550 | 0.935~0.975 |
| | Metabolic | 404 | 0.8632 | 0.828~0.895 |
| | PB | 3342 | 0.9322 | 0.924~0.941 |
| | King James | 1810 | 0.8877 | 0.873~0.902 |

We focused on the improvement of the algorithm in this paper, but in the future, we will also do further research on large networks because the large-scale network has more aspects it needs to consider, such as the average path length of the network, clustering coefficients of large networks and other important network indicators. Many large-scale networks have special forms of clustering coefficients, although the degree varies. Li et al. [32] showed that the average clustering coefficient of the network with large degree accords with asymptotic expression. This provides a new research guide for our next research work.

For large-scale network, in theory, better results should be achieved because the more data, the more references and the more fully trained the network model is, so better results can be achieved, but the parameters in the network are also very important. At the same time, the larger the network, the longer the training time, therefore the amount of resources needed to calculate will also increase, and these are factors that we will need to consider.

### 5.3.3. Parameter Sensitivity

As previously stated, we concatenated the local and global structural feature vector into node information matrices for link prediction. The effect of different matrix sizes on data sets in our method is shown in Figures 6 and 7.

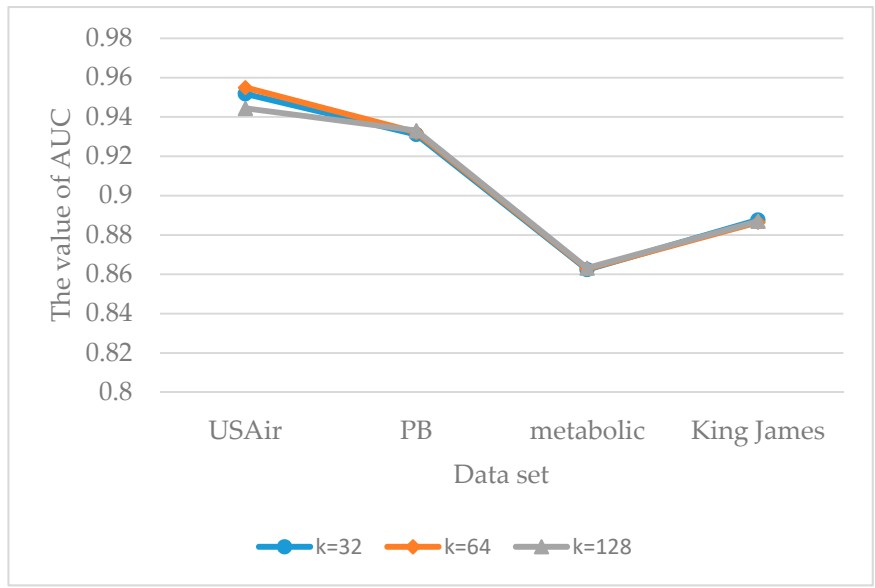

**Figure 7.** The effect of AUC value on model AM-ResNet with parameter *k*.

In Figure 6, the effect of different matrix sizes on the prediction results of the DenseNet model is investigated. As shown in Figure 6, the best size of the node information matrices is 64. When *k* is larger than 64, irrelevant nodes are learned by DenseNet. In other words, each node corresponding

to the sub-graphs which are generated by Algorithm 1 encloses a set of nodes that are not related to the source node. Unrelated information thereby is integrated into node information matrices, which leads to the situation where the prediction efficiency of the model is slightly worse. Similarly, if the number of nodes in the sub-graph is insufficient, a number of existing links in the network would not be predicted. The best size in Figure 7 is also 64. However, no matter which value of *k* is taken, the overall difference among the three is not significant. The reason is that the attention mechanism introduced in AM-ResNet model can grasp the main characteristics of the link structure. Therefore, no matter which value *k* takes, its prediction effect is not very different.

As can be seen in Figures 6 and 7, the overall effect of DenseNet model is better than that of AM-ResNet model. The reason was mentioned in Section 4.7, that DenseNet model is a direct connection of feature maps from different layers which leads to the situation that the transmission efficiency of information flow is better than that of AM-ResNet, but its stability is not as good as AM-ResNet.

## 6. Conclusions

Recently, link prediction has attracted more attention of researchers in different disciplines, and various link prediction algorithms are stacked, which has advantages and disadvantages. The existing link prediction methods based on similarity cannot better express some non-linear modes which play a decisive role in the link in the network because they are designed manually. Although the link prediction method based on the shallow neural network can make good use of the potential features of the network nodes, it cannot capture the deep non-linear features, such as link structure features. Deep neural networks, such as deep convolutional neural networks, can capture deep non-linear features and learn more useful features because of their strong feature learning ability, and thus improve the accuracy of classification or prediction. Therefore, we use the deep convolutional neural network to predict the link. Firstly, the link in the network is treated as matrix pairs, and then the structural similarity of links is captured by deep convolutional neural network. Finally, the experimental results show that the method proposed in this paper is better than the common benchmark algorithm. There are still some challenges ahead.

(1) We considered the pure structure of the network for link prediction in this paper, although it can achieve better prediction results on most networks, but this is obviously not enough. Therefore, in the future, other information will be added to the network for modeling, in order to achieve better prediction results.

(2) Modeling with deep convolution neural network can improve the effect of link prediction, but it is well known that the computational complexity will also increase. Therefore, the use of the in-depth learning model for large-scale network data is bound to be constrained by efficiency, so how to improve the efficiency of the model is also a challenge at present.

(3) Network adaptability. In this paper, we adopted some common data sets, which cannot be tested on large-scale networks, so we will consider this research step by step.

**Author Contributions:** W.W. and Y.H. discussed and confirmed the idea; Y.H. and L.W. carried out the experiment and analyzed the data; L.W. wrote the paper; H.W. reviewed the paper; R.Z. carried out project administration.

**Funding:** This research was funded by The National Natural Science Foundation of China (61772562) and "The Fundamental Research Funds for the Central Universities", South-Central University for Nationalities (CZY18014) and Innovative research program for graduates of South-Central University for Nationalities (2019sycxjj120).

**Conflicts of Interest:** The authors declare no conflict of interest.

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
