# Peer review of "Link Prediction Based on Deep Convolutional Neural Network"

_information, doi:10.3390/info10050172_

Round 1

Reviewer 1 Report

The paper proposes advances in the use of deep learning techniques for link prediction in networks. The results show some improvements over previous standard techniques as well as over analogous deep learning attempts. The design of the experiments seems sound, however I think the authors should put an additional effort to compute confidence intervals around the validation scores they propose: as these are the result of random experiments, it would be enough to repeat the experiments and give an idea of the distribution of the results for the various methods.

My main concern with the paper, however, is the poor exposition of the methods. Throughout the paper there is a need for a proper revision of the exposition, as several sentences are very hard to grasp. Some concepts or ideas of the paper are only vaguely explained (e.g. lines 106-108) and can only be understood by reading other papers. Moreover some quantities are not defined at all (e.g. What is the growth rate 'k' in the label of fig. 3?). Another example is the procedure for subgraph extraction that is only described through a piece of pseudo-code, but not described nor motivated in the text.

Author Response

Dear  Reviewers:

Thank you for your letter and for the reviewers’ comments concerning our manuscript entitled Link Prediction Based on Deep Convolutional Neural Network(ID: information-464423). Those comments are all valuable and very helpful for revising and improving our paper, as well as the important guiding significance to our researches. We have studied comments carefully and have made correction which we hope meet with approval. Revised portion are marked in red in the paper. The main corrections in the paper and the responds to the reviewer’s comments are as flowing:

Point 1: The design of the experiments seems sound, however I think the authors should put an additional effort to compute confidence intervals around the validation scores they propose: as these are the result of random experiments, it would be enough to repeat the experiments and give an idea of the distribution of the results for the various methods.

Response:

Considering the Reviewer’s suggestion, we have made the following modifications:

In order to further illustrate the experimental results, we calculate the confidence interval of the classification accuracy predicted by the proposed method. Because we regard link prediction as a binary decision, which may be correct or wrong, we assume that it is general. We are interested in 95% confidence interval. The formula is as follows:

Pr(c1<=μ<=c2)=1-α

Where α is the significant level (we choose 0.05), μ represents the average value of AUC, Pr is the abbreviation of probability.

Model

Dataset

Simple   size

AUC

interval(c1,c2)

DenseNet-LP

USAir

424

0.9554

0.935~0.975

Metabolic

404

0.8742

0.836~0.902

PB

3342

0.9474

0.940~0.955

King   James

1810

0.8966

0.882~0.910

AM-ResNet-LP

USAir

424

0.9550

0.935~0.975

Metabolic

404

0.8632

0.828~0.895

PB

3342

0.9322

0.924~0.941

King   James

1810

0.8877

0.873~0.902

We calculate the confidence intervals of the predicted results of the two methods, in which the sample we take is the size of the test data set. As can be seen from Table 5, the confidence interval ranges from 1.5% to 7%, which shows that our experimental results are credible. (Page 12-13, line 442-455);

Point 2: My main concern with the paper, however, is the poor exposition of the methods. Throughout the paper there is a need for a proper revision of the exposition, as several sentences are very hard to grasp. Some concepts or ideas of the paper are only vaguely explained (e.g. lines 106-108) and can only be understood by reading other papers. Moreover some quantities are not defined at all (e.g.What is the growth rate 'k' in the label of fig. 3?). Another example is the procedure for subgraph extraction that is only described through a piece of pseudo-code, but not described nor motivated in the text.

Response:

Some concepts of this paper have been further explained, and the ideas of this paper have been further elaborated.

e.g. lines 106-108?

Response:

What I'm trying to say here is: Because the output of the network front layer. (Page 3, line 117-118)

e.g. What is the growth rate 'k' in the label of fig. 3?

Response:

We are very sorry for our negligence of K’s explanation.

K denotes the number of feature map output from each layer of the network. In this diagram, we draw three. (Page 5, line 186-187)

e.g. Another example is the procedure for subgraph extraction that is only described through a piece of pseudo-code, but not described nor motivated in the text.

Response:

A significant limitation of link prediction heuristics is that they are all handcrafted features, which have limited expressibility. Thus, they may fail to express some complex nonlinear patterns in the graph which actually determine the link formations [19]. (Page 6, line 223-225)

We tried our best to improve the manuscript and made some changes in the manuscript.  These changes will not influence the content and framework of the paper. And here we did not list the changes but marked in red in revised paper.

We appreciate for Reviewers’ warm work earnestly, and hope that the correction will meet with approval.

Once again, thank you very much for your comments and suggestions.

Reviewer 2 Report

I read this paper with interest. This paper uses the deep convolutional neural network to predict the link. The link in the network are treated as matrix pairs, and then the structure similarity of links is captured by deep convolutional neural network. The experimental results show that the method proposed in this paper is better than the common benchmark algorithm. The results are new as far as I am concerned. The presentation needs some improvement. The paper can be accepted if the following comments are carefully taken into consideration. I will give my recommendation based on the revised version. The detailed comments are as follows.

(1) Line 24, "It construct a model of the ...." should be "It constructs a model ...".
(2) Line 25, why the S in "Sub-graph" is capitalized? Does this refer to something special?
(3) In the second paragraph on page 2, another important reason that structure-based method is not that reliable is that different networks may have very similar topological measures. A seminal work is: Distinct clusterings and characteristic path lengths in dynamic small-world networks with identical limit degree distribution. This aspect should be remarked.
(4) Line 93, it is not clear to me as to what is "a low rate of information transmission".
(5) In general, I feel the contribution and novelty can be better highlighted in Introduction situated in the context of existing work. For example, the three points stated between lines 86 and 94 should be referred to some existing work mentioned in Section 2. A certain degree of novelty is certainly essential for a paper to be published in Information.
(6) In line 106, it is not clear what the authors are going to say by "the data of several layers in the front of the network". Please re-phrase the sentence.
(7) Line 134, "node2vec has achieve very good...", here "achieve" should be "achieved". There are some other grammar errors and clerical errors in the paper. Please double check carefully.
(8) Equation (1) should be explained. What is the point of taking logarithm?
(9) In line 188, what is A_{ij}? I suppose they are the elements of the adjacency matrix A. Am I right? But this is not mentioned in the paper. For an academic paper, all relevant definitions should be provided so as to render a self-contained work. Please check other places in the paper.
(10) In the subgraph extraction part, it should be commented that special attention should be paid to the different methods of subgraph extractions as highlighted in the seminal work (in terms of robustness): Subgraph robustness of complex networks under attacks.
(11) Line 216, there is a blank square in the notation "G(  )". Please correct.
(12) The motivation of Definition 2 should be explained. Eq. (2) seems to be enigmatic.
(13) Line 296, I do not follow why this "may hinder the effective transmission of information flow...". An explanation is appreciated.
(14) Between lines 386-394, it should be noted that many large-scale networks have special form of clustering coefficients (see the recent work: Clustering coefficients of large networks). These networks are typically much larger than the four networks considered in this paper. What would you expect the results would be? A brief discussion is needed.
(15) The conclusion section can be improved by adding more open problems, which are not necessarily what you are going to do but can also be something challenging. This will be very helpful for interested readers of Information.

Author Response

Dear  Reviewers:

Thank you for your letter and for the reviewers’ comments concerning our manuscript entitled Link Prediction Based on Deep Convolutional Neural Network(ID: information-464423). Those comments are all valuable and very helpful for revising and improving our paper, as well as the important guiding significance to our researches. We have studied comments carefully and have made correction which we hope meet with approval. Revised portion are marked in red in the paper. The main corrections in the paper and the responds to the reviewer’s comments are as flowing:

Point 1: Line 24, "It construct a model of the ...." should be "It constructs a model ..."

Response:

Thanks very much for pointing the English written issue, and we have re-check the entire paper and revised or improved several English written issues (Page 1, line 24).

Point 2: Line 25, why the S in "Sub-graph" is capitalized? Does this refer to something special?

Response:

The initials are capitalized for convenience because we prompted for syntax errors when writing in word documents, so we changed lowercase to uppercase. (Page 1, line 25).

Point 3: In the second paragraph on page 2, another important reason that structure-based method is not that reliable is that different networks may have very similar topological measures. A seminal work is: Distinct clusterings and characteristic path lengths in dynamic small-world networks with identical limit degree distribution. This aspect should be remarked.

Response:

It is really true as Reviewer suggested that another important reason we ignored, after referring to “Distinct Clusterings and Characteristic Path Lengths in Dynamic Small-World Networks with Indentical Limit Degree Distribution”, we made the following corrections.

Furthermore, structure-based method is not that reliable, there is such a situation, different networks may have distinct clusterings and path length but have the same degree distributions. (Page 2, line 56-57)

Point 4: Line 93, it is not clear to me as to what is "a low rate of information transmission".

Response:

We have rewritten it, and we have explained this in Chapter 4.7. (Page 2, line 94-95)

Point 5: In general, I feel the contribution and novelty can be better highlighted in Introduction situated in the context of existing work. For example, the three points stated between lines 86 and 94 should be referred to some existing work mentioned in Section 2. A certain degree of novelty is certainly essential for a paper to be published in Information.

Response:

We rewrote our contributions and innovations and highlighted our work in the introduction, as follows: (Page 2-3, Line 88-105)

(1) To solve the link prediction problem, we transform it into a binary classification problem and construct a deep convolution neural network model to solve the problem.

(2) In view of the fact that heuristic methods can only utilize the network's topological structure and represent learning methods can only utilize the potential features of the network, such as DeepWalk, LINE and node2vec. We propose a sub-graph extraction algorithm, which can better contain the information needed by the link prediction algorithm. On this basis, a residual attention model is proposed, which can effectively learn from graph structure features to link structure features.

(3) Through further research, we find that residual attention mechanism may hinder the transmission of information in the model. Therefore, a Densely Convolutional Neural Network model is proposed to improve the effect of link prediction.

Point 6: In line 106, it is not clear what the authors are going to say by "the data of several layers in the front of the network". Please re-phrase the sentence.

Response:

Thank you for pointing out the error. We have rewritten the statement. (Page 3, line 117-118)

Point 7: Line 134, "node2vec has achieve very good...", here "achieve" should be "achieved". There are some other grammar errors and clerical errors in the paper. Please double check carefully.

Response:

Thanks very much for pointing the English written issue, and we have re-check the entire paper and revised or improved English written issues. (Page 4, line 145);

Point 8: Equation (1) should be explained. What is the point of taking logarithm? 

Response:

This formula is an optimization function analogous to skip-gram model, in which log function is taken to simplify the operation. (Page 3, line 152)

Point 9: In line 188, what is Aij? I suppose they are the elements of the adjacency matrix A. Am I right? But this is not mentioned in the paper. For an academic paper, all relevant definitions should be provided so as to render a self-contained work. Please check other places in the paper.

Response:

You understand it very correctly. We have revised it in this article. (Page 7, line 278-279)

Point 10: In the subgraph extraction part, it should be commented that special attention should be paid to the different methods of subgraph extractions as highlighted in the seminal work (in terms of robustness): Subgraph robustness of complex networks under attacks. 

Response:

Thank you for your comments. In response to your comments, we have made the following modifications in the corresponding parts of the article.

Network robustness depends on several factors including network topology, attack mode, sampling method and the amount of data missing, generalizing some well-known robustness principles of complex networks [26]. Our proposed subgraph extraction algorithm is mainly used for sampling, so we discuss and care about the robustness of sampling.

Subgraph algorithm describes the “h-hop surrounding environment of node x, since G(·) contains all the nodes within h hops to x and corresponding edges. For example, when h>=2, the extracted subgraph will contain all the information needed to calculate and first-order and second heuristics algorithms, such as CN, PA and AA. However, this kind of algorithm only considers the topology of the second-order path, the time complexity is low, but the prediction effect is also poor. Therefore, the information learned from the subgraph is at least as good as most heuristics, and the robustness is also strong. (Page 6, line 234-244)

Point 11: Line 216, there is a blank square in the notation "G ( )". Please correct.

Response:

Thank you for your comments, we have made modification. (Page 6, line 232)

Point 12: The motivation of Definition 2 should be explained. Eq. (2) seems to be enigmatic. 

Response:

Motivation: However, for network topology data form, which is irregular and unordered, the convolution operator is ill-suited for leveraging spatially-local correlations in the network structure [27]. So inspired by this, the second step of the algorithm is to transform the local structure of each node into an ordered sequence. (Page 6, line 252-254)

For formula 2, we add the following explanation:

Where X, Y represent the representation vector of the node x, y, respectively, and the subscript i represents the dimension, for a total of d dimensions. The similarity is measured by calculating the cosine distance between them. The higher the vector similarity of the two nodes, the closer the two nodes are, so they should be in the first place when ranking. (Page 7, line 267-270)

Point 13: Line 296, I do not follow why this "may hinder the effective transmission of information flow...".An explanation is appreciated.

Response:

Because the original layer data is the best input for the next layer, instead of adding the original layer and the non-linear function mapping as the input for the next layer, which may impede the transmission of information in the network [23]. This is also the idea of dense convolution neural network.

Point 14: Between lines 386-394, it should be noted that many large-scale networks have special form of clustering coefficients (see the recent work: Clustering coefficients of large networks). These networks are typically much larger than the four networks considered in this paper. What would you expect the results would be? A brief discussion is needed.

Response:

This paper focuses on the improvement of the algorithm, not on adapting to different networks. By contrast, the standard data sets are common, and this article needs to be further studied on the large networks you mentioned.

Considering the Reviewer’s suggestion, we have made the following modification.

We focuses on the improvement of the algorithm in this paper, but in the future, we will also do further research on large networks because the large-scale network has more aspects it needs to consider, such as the average path length of the network, Clustering coefficients of large networks and other important network indicators. Many large-scale networks have special form of clustering coefficients, although the degree varies, Shang [30] et al. showed that the average clustering coefficient of the network with large degree accords with asymptotic expression. This provides a new research guide for our next research work.

For large-scale network, in theory, better results should be achieved because the more data, the more reference, the more fully trained the network model, so better results can be achieved, but the parameters in the network are also very important. At the same time, the larger the network, the longer the training time, the amount of resources needed to calculate will also increase, and these will be factors that we will need to weigh. (Page 13, line 456-467)

Point 15: The conclusion section can be improved by adding more open problems, which are not necessarily what you are going to do but can also be something challenging. This will be very helpful for interested readers of Information.

Response:

We have discussed the following in the conclusion section. (Page 14-15, Line 505-518)

(1) We considers the pure structure of the network for link prediction in this paper, although it can achieve better prediction results on most networks, but this is obviously not enough. Therefore, in the future, other information will be added to the network for modeling, in order to achieve better prediction results.

(2) Modeling with deep convolution neural network can improve the effect of link prediction, but it is well known that the computational complexity will also increase. Therefore, the use of in-depth learning model for large-scale network data is bound to be constrained by efficiency, so how to improve the efficiency of the model is also a challenge at present.

(3) Network adaptability. In this paper, we adopt some common data sets, which cannot be tested on large-scale networks, so we will consider this research step by step.

We tried our best to improve the manuscript and made some changes in the manuscript.  These changes will not influence the content and framework of the paper. And here we did not list the changes but marked in red in revised paper.

We appreciate for Reviewers’ warm work earnestly, and hope that the correction will meet with approval.

Once again, thank you very much for your comments and suggestions.

Round 2

Reviewer 2 Report

I appreciate the authors' careful revision. Most of my comments have been addressed. I recommend publication after a couple of minor corrections. In Ref. [14], author name should be displayed as Shang, Y. 2. Similar comment applies for Ref. [27].

Author Response

Dear Reviewers:

        Thank you for your letter and for the reviewers’ comments concerning our manuscript entitled “Link Prediction Based on Deep Convolutional Neural Network”(ID: information-464423). Those comments are all valuable and very helpful for revising and improving our paper, as well as the important guiding significance to our researches. We have studied comments carefully and have made correction which we hope meet with approval. Revised portion are marked in red in the paper. The main corrections in the paper and the responds to the reviewer’s comments are as flowing:

Point 1:In Ref. [14], author name should be displayed as Shang, Y. 2. Similar comment applies for Ref. [27].

Response:

        Amend the text in accordance with the suggestions you have given, as follows:

[14] Y.LS.; Shang. Distinct Clusterings and Characteristic Path Lengths in Dynamic Small-World Networks with Indentical Limit Degree Distribution. J. J STAT Phys, 2012, 149, 505-518(page 16,line 544)

[27] Y.L.; Shang. Subgraph Robustness of Complex Networks Under Attacks. J. IEEE T Syst Man Cy-S. 2019, 49, 821-833.(page 16,line 578)

[30] Y.S., Li; Y.L., Shang; Y.T., Yang. Clustering coefficients of large networks. J. Information Sciences, 2017, 382-383, 350-358.(page 16,line 583)

        We tried our best to improve the manuscript and made some changes in the manuscript. These changes will not influence the content and framework of the paper. And here we did not list the changes but marked in red in revised paper.

        We appreciate for Reviewers’ warm work earnestly, and hope that the correction will meet with approval.

        Once again, thank you very much for your comments and suggestions.

Information EISSN 2078-2489 Published by MDPI AG, Basel, Switzerland RSS E-Mail Table of Contents Alert
Back to Top